# Silicon Photonic Phase Shifters and Their Applications: A Review

**DOI:** 10.3390/mi13091509

**Published:** 2022-09-12

**Authors:** Haoyang Sun, Qifeng Qiao, Qingze Guan, Guangya Zhou

**Affiliations:** 1Department of Mechanical Engineering, National University of Singapore, Singapore 117575, Singapore; 2Center for Intelligent Sensors and MEMS (CISM), National University of Singapore, Singapore 117608, Singapore

**Keywords:** silicon photonics, phase shifters, MEMS, thermo-optics, free-carrier-depletion, photonic accelerator, on-chip spectrometer, neuromorphic computing

## Abstract

With the development of silicon photonics, dense photonic integrated circuits play a significant role in applications such as light detection and ranging systems, photonic computing accelerators, miniaturized spectrometers, and so on. Recently, extensive research work has been carried out on the phase shifter, which acts as the fundamental building block in the photonic integrated circuit. In this review, we overview different types of silicon photonic phase shifters, including micro-electro-mechanical systems (MEMS), thermo-optics, and free-carrier depletion types, highlighting the MEMS-based ones. The major working principles of these phase shifters are introduced and analyzed. Additionally, the related works are summarized and compared. Moreover, some emerging applications utilizing phase shifters are introduced, such as neuromorphic computing systems, photonic accelerators, multi-purpose processing cores, etc. Finally, a discussion on each kind of phase shifter is given based on the figures of merit.

## 1. Introduction

The past few decades have witnessed a huge growth in silicon photonics. Photonic integrated circuits (PICs) have been widely used and studied in areas such as telecommunications, lab-on-a-chip sensing, and quantum computing [1,2,3,4,5,6,7]. Benefitting from the broadband optical transparency (from 1.3 μm to 8 μm), high refractive index (*n* = 3.4757 at *λ* = 1550 nm, room temperature), and compatible manufacturing process with matured complementary metal–oxide semiconductor (CMOS) technologies [8,9,10], the silicon-on-insulator (SOI) substrate has become one of the most important platforms for on-chip PICs [11,12,13]. To meet the rapidly increasing demand for data communication, optical path routing, and optical signal modulation, passive and active optical components based on the SOI platform have been extensively studied in the past few decades [8,9,10,14,15,16,17]. Furthermore, the commercialization of silicon photonics has begun to take shape [18]. Some matured and advanced commercial foundries, such as the Advanced Micro Foundry (AMF) from Singapore, the American Institute for Manufacturing Integrated Photonics (AIM) from the United States, and the Interuniversity Microelectronics Centre (IMEC) from Belgium, have made great efforts and built promising PIC component libraries, including strip and rib waveguide, power splitter, grating coupler, waveguide crossing, directional coupler, micro-ring resonator, thermal-optical phase shifter, and so on. With intensive efforts, the propagation loss of silicon wire waveguide has been reduced to below 1.0 dB/cm by researchers, which paves the way to build large-scale PIC applications [19]. Moreover, packaging technology has been extensively explored [20,21,22,23], which leads to a high-level chip-scale integration including on-chip components such as photodetector (PD), modulator, laser source, and fiber-to-chip coupler.

With the development of dense PICs, effective and high-performance on-chip active components are urgently needed to realize complex on-chip functions. Phase shifters are one of the most important components in building PICs. A building block that offers two inputs and two outputs capable of power tuning and phase shifting can be considered a fundamental unit in large-scale PICs [24]. The phase shifter here refers to modulating the phase of the transmission wave only without changing the amplitude, where the power tuning function can be obtained by forming the interferometer based on phase shifters. Using enough amounts of such building blocks, an arbitrary linear optical system can be built. Using well-integrated phase shifters, researchers have reported various applications such as neuromorphic computing systems [25,26], optical phased arrays [27,28,29,30], light detection and ranging (LiDAR) systems [31,32], on-chip spectrometers [33,34,35], photonic accelerators [26,36,37], and so on. In this review paper, we focus on the recent progress of phase shifters on the SOI platform.

For silicon photonics, phase shifting mechanisms are mainly based on micro-electro-mechanical systems (MEMS), thermo-optics effects, and free-carrier-dispersion effects, to name a few (see Figure 1). MEMS is an effective modulation mechanism with low power consumption and optical insertion loss [38]. Its modulation speed is commonly limited by mechanical frequency. The thermo-optic mechanism could be realized by a simple fabrication process. Its moderate modulation efficiency and low insertion loss are preferred [39]. Considering the further dense integration, its heat dissipation and thermal crosstalk should be well engineered. In addition, a fast and effective phase shifter can be obtained using the free-carrier-dispersion mechanism, where the optical loss induced by the free-carrier absorption should be well controlled to scale it up [14]. On the other hand, due to the lack of second-order nonlinearity, common silicon-based materials usually exhibit negligible electro-optic (EO) effects. Silicon-based modulators that utilize EO effects require heterogeneous integration of other materials, such as lithium niobate (LiNbO_3_), graphene, etc. [40,41].

Here, we review silicon-based phase shifters with a focus on three common modulation mechanisms (MEMS, thermo-optics effects, and free-carrier-dispersion effects). Then, applications based on phase shifters are introduced, and the advantages and disadvantages of different modulation mechanisms are discussed. Some promising works of phase shifters based on other heterogeneous integrated materials can be found in [42,43].

In this review, we introduce and discuss the current progress of phase shifters on the SOI platform and its applications. Starting from the second section, we describe the basic theory of phase shift and the method for experimental characterization. In the third section, MEMS-based phase shifters are discussed in detail. Their operation principles and performances are introduced. Next, we briefly introduce thermo-optics and free-carrier-depletion-based phase shifters in the fourth and fifth sections. In the sixth section, some notable applications utilizing phase shifters are presented. The last section includes the discussion about silicon photonic phase shifters.

## 2. Methodology

### 2.1. Phase Shift Principle

The phase shift of the optical wave in the waveguide can be obtained by [44]:(1)$$\Delta \phi =\frac{\Delta n_{eff}2\pi \Delta L}{\lambda }$$
where Δ*n_eff_* is the change in effective refractive index, Δ*L* is the change in optical path length, and *λ* is the wavelength.

Generally, MEMS-based phase shifters change the effective refractive index (Δ*n_eff_*) of the optical mode by modifying the mode shape or perturbing the evanescent field, or change the optical path length (Δ*L*) by switching the optical route. The thermo-optics phase shifters modulate Δ*n_eff_* of the optical mode by changing the material refractive index through the on-chip heater. The free-carrier-depletion-based phase shifters modulate Δ*n_eff_* of the optical mode by changing the waveguide core material refractive index through the change in carrier concentration.

### 2.2. Experimental Characterization of Phase Shift

The phase shift can be extracted by embedding the phase shifter in an imbalanced Mach–Zehnder Interferometer (MZI) or a ring resonator. By applying DC voltages with different amplitudes, the value of phase shift can be extracted from the MZI optical transmission spectrum according to [45]:(2)$$\Delta \varphi =\frac{\left|\lambda \left(V_{0}\right)-\lambda \left(V\right)\right|}{FSR}$$
where *λ(V*_0_*)* is one of the MZI spectral dip wavelengths at the initial state without applied voltage, *λ(V)* is the same MZI spectral dip wavelength with applied voltage, and *FSR* is the free spectral range of the MZI spectrum. The unit of Δ*ϕ* is 2π.

The *n_eff_* can also be extracted from the resonance wavelength tuning of a ring resonator [46]:(3)$$\Delta n_{eff}=\frac{\Delta \lambda_{res}\cdot m}{L},\;m=1,\;2,\;3\ldots$$
where Δ*λ_res_* is the resonance wavelength tuning and *L* is the round-trip length.

## 3. MEMS-Based Phase Shifter

MEMS-enabled photonics refers to an industrial technology that integrates optical, electrical, and mechanical fields on the micro- and nanoscale. It modulates the optical mode in the waveguide to realize a variety of functions by MEMS actuation. This technology has flourished in the past few decades with the development of advanced silicon micro- and nano-fabrication technologies, and its applications have gradually evolved from free-space optics such as digital micromirror devices (DMD) to on-chip PICs [47,48]. Due to the excellent optical and mechanical properties of silicon, such as low optical absorption loss, low cost, matured fabrication technologies, reliable mechanical properties, and excellent ability to integrate electronic functions, the SOI substrate has become one of the most important platforms for on-chip MEMS applications. The driving mechanisms include electrostatic actuation, electrothermal actuation, piezoelectric actuation, and so on.

The MEMS-based phase shifter has attracted tremendous attention from worldwide researchers in academia and industry due to its high modulation efficiency, ultra-low power consumption, small footprint, and low insertion loss. In this section, we first introduce the figures of merit (FOMs) that are used to evaluate the MEMS-based phase shifter and discuss the outstanding works so far according to these FOMs.

### 3.1. FOMs for MEMS-Based Phase Shifter

#### 3.1.1. Half-Wave Voltage Length Product (*V_π_**·L**_π_*)

MEMS-based phase shifters are mainly driven by electrostatic actuation; hence, a potential difference needs to be applied between the movable and rigid parts. *V_π_**·L**_π_* refers to the voltage that needs to be applied on the phase shifter of length *L**_π_* to achieve π phase shift, which indicates the modulation efficiency and footprint of the device. By embedding a phase shifter into one arm of the MZI, *V_π_* can be obtained by measuring the DC voltage required to modulate the MZI transmission at an exact wavelength from minimum to maximum. Thus, a smaller *V_π_**⋅L**_π_* indicates a higher efficiency for phase shift.

#### 3.1.2. Insertion Loss (IL)

MEMS-based phase shifters usually modify or perturb the optical mode in the waveguide, which inevitably has an impact on the optical transmission. Therefore, the insertion loss here not only refers to the initial insertion loss in the unactuated state but also includes the extra loss induced by MEMS actuation.

For MEMS-based phase shifters driven by electrostatic actuation or electrothermal actuation, oxide-to-air transition loss happening in the rigid-to-movable region is another source, while piezoelectric actuation method does not suffer from this loss since the entire phase shifter area is wrapped in cladding material [49,50,51]. Additionally, in some cases, mode conversion is involved, such as ridge-to-slot transition, which induces extra loss.

For phase shifters embedded in an imbalanced MZI, the power imbalance between two branches of the MZI can be extracted by fitting the measured MZI transmission spectrum to the theoretical one [52].

#### 3.1.3. Response Time

As the dynamic response of the MEMS-based phase shifter is determined by its mechanical structure, mechanical frequency should be measured to evaluate its response time. The 3 dB cutoff bandwidth refers to the frequency of the dynamically modulated signal (AC voltage) applied when the modulated power amplitude variation of MZI is decreased by 3 dB. The mechanical resonant frequency (*f*) can be extracted from the 3 dB measurements, and the response time can be estimated to be 1/*f*. Specifically, the response time can be read from the optical output rise and fall time by applying a square wave modulated voltage signal.

### 3.2. Modulation Mechanism

In the literature, the modulation mechanisms of the MEMS-based phase shifter fall into three categories, as shown in Figure 2. Figure 2a,b shows the modulation mechanisms by perturbing the evanescent field of the optical mode in the bus waveguide through a mechanical beam with mode cut-off dimensions. Figure 2c,d shows the modulation mechanism by directly modifying the optical mode field distribution in the waveguide. The modulation mechanism shown in Figure 2e,f is to modulate the optical path length.

#### 3.2.1. Evanescent Field Perturbation

This type of phase shifter changes the effective refractive index of the optical mode by perturbing its evanescent field. The relevant working principles are shown in Figure 2a,b. The perturbation structure was designed to avoid mode coupling or leaking from the waveguide. In 2016, Pruessner et al. proposed a three-dimensional phase shifter configuration utilizing one silicon nitride bridge placed above the bus waveguide to perturb its optical evanescent field (Figure 3a,b) [53]. The 120 μm long phase shifter achieved π phase shift under 3.8 V applied voltage and 2π phase shift under 4.2 V applied voltage. However, the over-perturbation by the silicon nitride beam and the gold layer coated on it may induce large extra loss during modulation (1.5 dB extra loss after 2p phase shift). To overcome this limitation, the authors proposed to modulate the phase shifter through gradient electric fields instead and increased the initial gap between the silicon nitride beam without the gold layer and the bus waveguide. In this case, the 100 μm long phase shifter realized p phase shift with 33 V applied voltage, and the extra loss was reduced to 0.04 dB. Abdulla et al. placed a silicon cantilever above the ring resonator to perturb the evanescent field, as shown in Figure 3c [54]. Δ*n_eff_* was induced by moving down the silicon nitride cantilever and thus changing the optical mode of the ring resonator, leading to a Δ*λ* of resonance wavelength. The phase shifter showed large nonlinear relationship between phase shift and applied voltage. Additionally, 122 pm resonance wavelength tuning was observed with a modulation depth of 18 dB.

In addition to the vertical perturbation using a MEMS-tunable layer above the SOI wafer, the evanescent field perturbation can be obtained using the silicon device layer in the SOI wafer only by in-plane or out-of-plane MEMS actuation. For example, Errando-herranz et al. placed a narrow silicon beam on one side of the ring resonator to utilize a longer effective optical path length, as shown in Figure 3d [55]. The ring resonator acting as the MEMS cantilever was electrostatically actuated, which induced out-of-plane motion. By fully etching the silicon between the perturbation beam and ring resonator, the buried oxide (BOX) layer beneath the thin gap was exposed for HF wet etching. After sacrificing the BOX part below the thin gap, a movable cantilever region was formed. The length of the cantilever was determined by an array of release holes. The tunable ring resonator achieved resonance wavelength tuning of 530 pm with a power consumption less than 100 nW and a tuning rate of 62 pm/V. Furthermore, the phase shifter showed promising scalability because of the small footprint. In addition, M. Poot used the H-resonator actuator, placing the gold electrode away from the bus waveguide to reduce the extra insertion loss caused by the metal absorption (Figure 2e) [56]. More than 0.5p phase shift was achieved with a 170 μm long phase shifter under 5 V applied voltage. Edinger et al. utilized a comb drive actuator to modulate the *n_eff_* of the bus waveguide [45]. Compared with the parallel plate capacitor MEMS actuator, the comb drive actuator provided a larger displacement in a more accurate and stable manner by sacrificing footprint. A 17.2 μm long phase shifter achieved p phase shift with 10.7 V applied voltage, which showed a *V_π_**·L**_π_* of 0.0184 V⋅mm. Through balancing the resonant frequency and *V_π_*, the 3 dB cut-off bandwidth was measured as 503 kHz and the power consumption is 500 nW with p phase shift. It is noted that an approximately linear relationship between the phase shift and actuation voltage is achieved by optimizing the optical and mechanical design simultaneously, which could facilitate its further applications in the large-scale packaged PICs.

#### 3.2.2. Confined Optical Mode Modification

This type of phase modulator directly changes the optical mode field distribution by mechanically moving the waveguide structure, thereby changing the effective refractive index of the waveguide mode. In the literature, slot waveguide and directional coupler have mainly been adopted. In 2012, Acoleyen et al. presented a phase shifter by reducing the slot waveguide air gap, as shown in Figure 4a [57]. They applied a potential difference between two silicon arms of the slot waveguide, thereby changing the slot mode *n_eff_*. By cascading three 5.8 μm long tunable slot waveguides, the authors achieved 0.22p phase shift under 13 V applied voltage. Larger phase shift can be achieved using a longer tunable slot waveguide at a cost of optical loss. After that, Feng et al. built a physical model about the slot waveguide phase shifter, and theoretically analyzed the influence of Casimir force, optical force, and electrostatic force while modulating the slot waveguide [44]. The mechanical model and pull-in effect were analyzed in detail, as well. To improve the performance of slot waveguide phase shifter, Grottke et al. and Baghdadi et al. used asymmetric slot waveguide and double-slot waveguide, respectively, as shown in Figure 4b,c [58,59]. The parallel plate capacitor MEMS actuator was used to realize the in-plane motion of the two silicon nitride beams of the slot waveguide. Instead of reducing the air gap, Grottke et al. deposited a gold electrode near one side of the slot waveguide and increased the air gap by applying a bias voltage between the gold electrode and one beam of slot waveguides. In this study, a 250 μm long phase shifter was fabricated and achieved a *V_π_* of 4.5 V and a phase shift of 13p at 17 V applied voltage. In addition, they used an asymmetric slot waveguide to suppress the generation of higher-order eigenmodes to reduce insertion loss. The static insertion loss of the 250 μm long phase shifter is lower than 0.7 dB, and the resonant frequency was measured as 779 kHz in vacuum. Baghdadi et al. used dual-slot waveguide to improve modulation efficiency and achieved p phase shift with 25 μm long phase shifter under 0.85 V applied voltage. An insertion loss less than 0.04 dB was extracted from the measured MZI transmission spectrum, and the 3 dB cutoff bandwidth was approximately 0.26 MHz.

In addition, Sattari et al. and Liu et al. almost simultaneously proposed a phase shifter via modulating the vertical directional coupler supermode effective refractive index [60,61]. Sattari et al. investigated the performance of the phase shifter and proposed a MEMS actuator design with two step actuations, as shown in Figure 4d. A stopper was designed to prevent the pull-in effect. In addition, Liu et al. experimentally demonstrated a 150 μm long phase shifter on an indium phosphide membrane on silicon, as shown in Figure 4e. A *V_π_**·L**_π_* of 0.58 V mm was achieved, and a 4 dB extra loss was induced during the modulation.

#### 3.2.3. Optical Path Length Adjustment

The working principle is to change the phase of the bus waveguide transmission wave by adjusting the optical path length. Chiu et al. proposed to adjust the optical path length by bending one long and suspended waveguide, as shown in Figure 5a [62]. The authors applied a bias voltage between the suspended waveguide and the electrode to deform the waveguide. Experimental results found that a 150 μm long phase shifter achieved 0.06p phase shift at a voltage of 200 V. It was found that the limited phase shift could be attributed to the small mechanical deformation. Moreover, Ikeda et al. realized an adjustable optical path length by integrating a movable waveguide region with a comb drive actuator (Figure 5b) [63]. Two directional couplers were designed to transfer the light from the input waveguide to the movable waveguide and out to the output waveguide in the following propagation. The phase shifter achieved 3p phase shift under the 13 V actuation voltage, and the displacement of the directional couplers was 1 μm. It should be noted that the proposed approach could be advantageous in terms of insertion loss owing to the well-maintained mode propagation during MEMS tuning.

### 3.3. Discussion

In this section, we introduced three MEMS-based phase shifter working mechanisms and the type of MEMS actuator used in their works in detail. Performances of some typical MEMS-based phase shifters are summarized in Table 1. MEMS-based phase shifters showed advantages of high efficiency, low insertion loss, and broad bandwidth. The modulation speed ranges from several hundred kHz to a few MHz. In MEMS-based phase shifters, most of them use electrostatic MEMS actuators. Hence, the modulation speed and required voltage are both affected by the size and type of the MEMS actuators. The modulation speed could be increased by designing the mechanical structure with a larger stiffness. However, this could lead to a larger electrostatic actuation voltage. Therefore, a trade-off between these two FOMs should be carefully considered for the application scenarios. For the evanescent field perturbation phase shifter, the phase modulation efficiency could be improved by placing the perturbation beam closer to the bus waveguide, but this incurs a larger optical loss at the same time. It is necessary to carefully determine the initial position and the width of the bus waveguide to balance the modulation efficiency and insertion loss. In addition, the modulation relationship between phase shift and voltage for the MEMS-based phase shifters are usually nonlinear, and the pull-in effect must always be avoided during modulation. For the non-solid-state system, reliability is an important factor that must be investigated due to the inherent failure risks such as fatigue and stiction. The failure mechanisms in MEMS devices have been widely studied [64]. Recently, Seok et al. explored the long-term reliability of a MEMS-actuated vertical coupler used in an optical switch, which showed negligible performance degradation after 10 billion times of actuation [65]. The packaging and integration with the existing silicon photonic platform need to be further studied, as well [21].

## 4. Thermo-Optics Phase Shifter

Thermo-optics phase shifters are widely adopted owing to their simple fabrication process, efficient phase shift modulation, and broad bandwidth. The thermo-optics coefficient is defined as the refractive index of the material to the change in the temperature (dn/dT), which is 1.87 × 10^−4^ at the wavelength of 1550 nm for silicon [66].

In this section, we first introduce a basic configuration and its working principle for the thermo-optics phase shifter. Next, some FOMs are presented, and optimizations method are discussed based on these FOMs.

### 4.1. Working Principle of Thermo-Optics Phase Shifter

The working principle of the thermo-optics phase shifter is to change the refractive index of the waveguide and cladding material by injecting a current into a resistive heater along them, thereby changing the effective refractive index of the optical mode. The relationship between the phase change and the temperature change is given as [39]:(4)$$\Delta \varphi \left(\Delta T\right)=\frac{2\pi }{\lambda }{\left(\frac{dn}{dT}\right)}_{eff}\Delta TL$$
where *λ* is the wavelength, and ${\left(\frac{dn}{dT}\right)}_{eff}$ is the change in the effective refractive index of the transmission mode versus the change in temperature. This coefficient is not only affected by the change in the refractive index of silicon, but also the change in the refractive index of the surrounding claddings (e.g., silicon dioxide, silicon nitride). Δ*T* is the change in the temperature, and *L* is the length of the heating waveguide region.

According to Equation (4), the required temperature change to achieve p phase shift is:(5)$$\Delta T_{\pi }=\frac{\lambda }{2\cdot L\cdot {\left(\frac{dn}{dT}\right)}_{eff}}$$

Thus, one of the FOMs, the power consumption, can be approximately given by [67]:(6)$$\Delta P_{\pi }=\Delta T_{\pi }\cdot G$$
where *G* is the thermal conductance between the heated waveguide and the heat sink in a unit of W/K.

Two other important figures of merit are the propagation loss of the waveguide and the modulation speed. The modulation speed can be evaluated by a time constant, which is determined by [67]:(7)$$\tau =\frac{H}{G}$$
where *H* is the heat capacity of the heated arm.

The gap of finite thermal conductance between the heat source and the waveguide is not considered in the above equations.

A common configuration of the thermo-optics phase shifter is shown in Figure 6a. The silicon waveguide is patterned in the cladding and a heater is placed above the waveguide. It is noted that the vertical gap between the heater and waveguide should be kept large enough to avoid excessive optical insertion loss. Hence, an upper cladding is usually grown and covers the silicon waveguide to isolate and support the metal heater. While designing a thermo-optics phase shifter, the width of the silicon, and the thickness and type of the cladding and heater must be carefully designed. The steady-state heat distribution for a conventional thermo-optics phase shifter with different kinds of claddings is shown in Figure 7a [68].

Based on the traditional thermo-optics phase shifter, many research efforts have focused on optimization targeting the power consumption, modulation speed, and insertion loss, as shown in Figure 6. Figure 6b achieves thermal insulation between the silicon waveguide and the claddings and substrate layer by processing a free-standing waveguide to improve power consumption. Figure 6c shows the method by reducing the vertical gap between the heater and the bus waveguide to improve power consumption and modulation speed. An optical transparent material (e.g., 2D material) is needed to prevent large propagation loss. Doping silicon can be used as the heater as well, as shown in Figure 6d,e, which shows adequate balance between these three FOMs. Detailed works based on these configurations are discussed in the following sections.

### 4.2. Typical Work in Thermo-Optics Phase Shifter

#### 4.2.1. Toward Low Power Consumption

One of the approaches towards a low-power-consumption thermo-optics phase shifter is to reduce the waste heat to the surrounding material other than the waveguide itself. Sun et al. proposed a 100 μm long free-standing waveguide thermo-optics phase shifter with a *P**_p_* of 540 μW, as shown in Figure 7b [70]. The insertion loss for an MZI switch that contains two proposed phase shifters was measured as 2.8 dB. However, the modulation speed degraded to less than 10 kHz due to slower heat dissipation.

Instead of reducing the heat dissipation, one can take advantage of power multiplexing to improve it. Benefitting from the spiral waveguide photonic structure, the heat generated by heaters can be absorbed almost entirely by the optical waveguide. While designing the layout of spiral waveguide, some interesting and effective methods were proposed to reduce the device propagation loss. Qiu et al. proposed to set the adjacent waveguide widths as different to reduce coupling loss and an offset at the connection part between the bending waveguide and the straight waveguide reduce mode mismatch (Figure 7c) [71]. The insertion loss for the phase shifters decreased from 1.9 dB to 0.9 dB after these optimizations, and power consumption reduced to 3 mW without sacrificing modulation speed (a modulation bandwidth of 39 kHz).

The third optimizing strategy is reducing the gap between the heater and the waveguide. A metal heater with smaller gap between the bus waveguide will improve power consumption and modulation speed while inducing a larger scattering loss. To solve this problem, some optical transparent materials with relatively high electric resistance were utilized, such as indium tin oxide (ITO) and graphene [68]. Yan et al. utilized a slow-light-enhanced silicon photonic crystal waveguide with graphene heaters deposited on it (Figure 7d) [69]. A tuning efficiency of 1.07 nm/mW and power consumption per free spectral range of 3.99 mW/FSR were achieved. The response time 750 ns was obtained.

In addition to the optimization of heating efficiency, the phase shifter could be advanced by multi-mode waveguide circuit design. Miller et al. proposed a method which used multiple direction coupler mode converters to route the multi-mode optical wave propagation [72]. The proposed approach effectively increased the heating optical path length by letting optical wave multi-pass the phase shift region, thus improving the modulation efficiency and reducing the power consumption (Figure 7e). By utilizing six mode converters, an 8-fold longer optical path extension could be achieved. They demonstrated only 1.7 mW *P**_p_* with a modulation speed of 6.5 μs. Compared with the widely adopted ring resonator type of phase shifter, the proposed approach could have a superior working bandwidth with the optical path extension. The insertion loss reached up to 6 dB due to the cascading of multiple mode converters and could be improved by optimizing the optical structure and fabrication process.

#### 4.2.2. Toward Low Loss and High Modulation Speed

A suitable balance between modulation speed, power consumption, and propagation loss could be achieved by doping the same carrier on both sides of the waveguide. The heat is generated by applying a continuous current to the doped part, and its steady-state heat distribution is shown in Figure 8a [67]. A 357 kHz modulation bandwidth could be achieved by improving the proximity of the heat source and the waveguide [73]. A more compact design decreased the heated arm heat capacity *H* and the time constant *τ*. At the same time, by carefully designing the doping silicon distribution, the insertion loss of the phase shifter could be significantly reduced to 0.23 dB for a 61.6 μm long phase shifter [74]. Some typical thermo-optics phase shifters based on doping silicon heaters are shown in Figure 8b,c [75].

### 4.3. Discussion

Some typical thermo-optics phase shifters are summarized in Table 2. Compared with the phase shifter without air trenches, the one with air trenches shows much less power consumption but has a slower modulation speed. While designing and using a thermo-optics phase shifter, a trade-off between power consumption and modulation speed is an important factor for researchers to consider, as thermal inductance has opposite effects on these two FOMs. Thus, a composite FOM *P**_π_*·*τ* is widely used to characterize a thermo-optics phase shifter that relies on the designed thermal inductance.

In addition to the structure of the thermo-optics phase shifter, the type of heater is also a factor to consider. Commonly used heaters include metal, which is placed on top of the bus waveguide, and a doped-silicon resistor, which is placed on both sides of the bus waveguide. According to the experimental results of thermo-optics phase shifters with different types of heaters processed in the commercial foundries IMEC and AMF, phase shifters with metal heaters and doped-silicon heaters show similar modulation efficiencies, while the modulation speed of doped-silicon-based phase shifters is faster but has a larger footprint [67,76]. Besides benefitting from the excellent optical properties and high electric resistance, optical transparent materials such as graphene and ITO are also favorable candidates for heater materials.

Even with a moderate modulation speed, the thermo-optics phase shifter is widely preferred in silicon photonics due to its high modulation efficiency and easily access from commercial foundries. Considering a densely integrated on-chip system, a thermo-optics phase shifter requires not only a calibration of the initial state, but also avoiding thermal crosstalk by planning the layout. Random phase noise is another factor that needs to be considered in some exact applications. Song et al. demonstrated a 2 μm width silicon photonic thermo-optics phase shifter with a TiN heater, which reduced the normalized phase error to 1e-3 π/nm [77].

**Table 2 micromachines-13-01509-t002:** Performance comparison of silicon photonic thermo-optics phase shifters.

	Heater Type	Waveguide Type	Power Consumption (mW)	Modulation Speed (μs)	Insertion Loss (dB)	Ref.
Conventional phase shifters	Tungsten	Strip waveguide	23.38	45	-	[78]
	TiN	Strip waveguide	21.4	5.6	<0.01	[67]
Optical transparent heater	ITO	Strip waveguide	10	5.2	<0.01	[68]
	Graphene	Rib waveguide	57.75	4.97	2	[79]
Air-trenches phase shifter	Pt	Strip waveguide	0.54	141	2.8	[70]
	TiN	Strip waveguide	0.49	144	0.3	[80]
Doped silicon	Doped silicon	Bend waveguide	12.7	2.4	0.5	[75]
	Doped silicon	Rib waveguide	24.77	7.7	0.23	[74]
	Doped silicon	Rib waveguide	22.8	2.2	<0.01	[67]
	NiSi	Rib waveguide	20	2.8	-	[73]
Spiral waveguide	Cr/Au	Strip waveguide	6.5	14	-	[81]
	Ti	Strip waveguide	3	25.64	0.9	[71]

## 5. Free-Carrier-Depletion-Based Phase Shifter

Free-carrier-dispersion-based phase shifters are favored in the field of telecommunications and data centers due to their high modulation speed and low power consumption. Based on the working mechanism, free-carrier-dispersion-based phase shifters fall into three categories: carrier injection, carrier depletion, and carrier accumulation. In this section, we introduce the free-carrier-depletion-based phase shifter only. Some outstanding reviews of free-carrier-based phase shifters can be found in [9,14,15,82].

### 5.1. Modulation Principle

Free-carrier-depletion-based phase shifters usually modulate the phase of transmission wave by changing the carrier concentration in the core material of the bus waveguide. The refractive index changes (Δ*n*) and carrier absorption (Δ*α*) caused by free-carrier concentration change can be described by the Drude model [83]:(8)$$\Delta n=-\frac{e^{2}\lambda^{2}}{8\pi^{2}c^{2}\varepsilon_{0}n}\cdot \left(\frac{\Delta N_{e}}{m_{e}^{*}}+\frac{\Delta N_{h}}{m_{h}^{*}}\right)$$
and
(9)$$\Delta \alpha =\frac{e^{3}\lambda^{3}}{16\pi^{3}c^{3}\varepsilon_{0}n}\cdot \left(\frac{\Delta N_{e}}{m_{e}^{*2}\mu_{e}}+\frac{\Delta N_{h}}{m_{h}^{*2}\mu_{h}}\right)$$
where *e* refers to the elementary charge, λ is the laser wavelength, *c* is the light speed, $\varepsilon_{0}$ denotes the vacuum permittivity, *n* represents the unperturbed refractive index of the material, Δ*N* is the charge carrier density, *m** refers to the carrier effective mass, and the subscripts *e* and *h* indicate quantities related to electrons and holes, respectively.

Some free-carrier-depletion-based phase shifter structures are shown in Figure 9. Rib waveguide is usually used, benefitting from a pair of thin film slabs. The cross-section is divided into an enhanced doping concentration region (p++/n++ region), a doping concentration region (p+/n+ region), and an intrinsic region (i region). Doping area distributions and doping concentrations are the most important parameters, which affect the modulation efficiency (*V_π_**·L**_π_*) and waveguide propagation loss (*α*). Electro-optic bandwidth is regarded to represent the modulation speed.

### 5.2. Typical Work in Free-Carrier-Depletion-Based Phase Shifter

Various configurations have been proposed to balance and optimize modulation efficiency, waveguide propagation loss, and modulation speed. The waveguide propagation loss can be effectively reduced by avoiding the overlap between the waveguide mode field and the doping area. Patel et al. proposed a phase shifter with an offset doping area, which aims at reducing the optical loss and improving the modulation efficiency [84]. The target doping concentration of the p type region was 7.8e17, which is lower than that of the n type region (2.1e18). The insertion loss of the 500 μm long doping waveguide embedded in a Michelson interferometric modulator was characterized as 4.7 dB, and a 0.72 V·cm *V_π_**·L**_π_* was obtained at 1V bias voltage. Figure 9b shows a PIPIN diode phase shifter proposed by Ziebell et al. [85]. By selectively doping the waveguide (8e17 in the p+ region, 1e18 in the n+ region, and 3e17 in the p region), the transmission loss was reduced while ensuring effective modulation efficiency and modulation speed. The experimental results showed that for a 0.95 mm long phase shifter embedded in the MZI, the insertion loss was extracted as 2.5 dB, and the *V_π_**·L**_π_* was 3.5 V·cm. The modulation speed was measured as 40 GHz. Tu et al. demonstrated the carrier compensation method and set the concentration of the doped waveguide at the corner to zero [86], thereby reducing the waveguide propagation loss to 1.04 dB/mm without sacrificing the modulation efficiency (Figure 9c). The *V_π_**·L**_π_* was measured as 2.67 V·cm at 6 V bias voltage. Azadeh et al. constructed a silicon–insulator–silicon capacitive phase shifter that greatly reduced the doped waveguide area, as shown in Figure 9d [87]. Through injecting a high concentration of carriers (7e18 in the n+ region and 6e18 in the p+ region), the waveguide propagation loss was obtained as 4.2 dB/mm with the modulation efficiency of 0.74 V·cm at 2V bias voltage. The modulation speed was measured as 48 GHz.

In addition, Li et al. proposed an ultra-fast free-carrier-deletion-based phase shifter by removing the silicon substrate beneath the bus waveguide, which can reduce the useless power consumption in the substrate and thus improve the modulation bandwidth (Figure 9e) [88]. The 3 dB EO bandwidth reached up to 60 GHz at the DC bias voltage of −8 V. The waveguide propagation loss was 2.2 dB/mm and the modulation efficiency achieved was 1.4 V·cm.

By maximizing the overlap between the depletion region and the optical mode, the modulation efficiency can be improved. As shown in Figure 9f,g, interleaved junctions and zig-zag structures were proposed, which demonstrated modulation efficiency of 2.4 V·cm and 1.7 V·cm, respectively [89,90].

### 5.3. Discussion

Performances of some typical free-carrier-depletion-based phase shifters are summarized in Table 3. Through optimizing the concentration and distribution of free carriers in the bus waveguide, research has been carried out to balance the modulation efficiency, modulation speed, and propagation loss. Phase shifters are widely used in the data transmission and telecommunication fields, benefiting from the fast modulation speed.

## 6. Other Phase Shift Modulation Mechanisms

Apart from the three phase shift modulation mechanisms mentioned above, two more modulation mechanisms (liquid crystal-based phase shifters and phase change materials) that utilize non-silicon-based materials but are still important are briefly introduced.

### 6.1. Liquid Crystal-Based Phase Shifter

The modulation efficiency of EO modulation directly on the silicon material is very low due to the weak second-order nonlinearity of silicon itself [104]. Benefitting from the high birefringence, liquid crystal material (e.g., E7 liquid crystal mixture) is a promising candidate to achieve EO modulation by injecting it above the silicon waveguide as cladding [105]. When no external electric field is applied, the director (the average orientation of the molecules) of the liquid crystal is parallel to the waveguide. In contrast, the director rotates, and its orientation becomes perpendicular to the waveguide while applying a sufficient large electric field. During the rotation process, the waveguide mode *n_eff_* is modulated, thus changing the phase of the transmission wave.

The commonly used waveguide platforms for liquid crystal-based phase shifters include strip waveguide platform and slot waveguide platform. Strip waveguide is easy to process, and the propagation loss can be maintained at a very low level. However, its modulation efficiency is relatively low due to the less evanescent field overlap with the liquid crystal claddings. On the other hand, a large portion of the optical field of slot mode is confined in the slot structure, which indicates large overlap between the optical field and liquid crystal claddings. In this case, the modulation efficiency is much larger and a larger propagation loss is induced.

Some promising works about liquid crystal-based phase shifters were proposed [106,107,108,109]. Atsumi et al. proposed a liquid crystal-based phase shifter utilizing strip waveguide [110]. By embedding it into a Michelson interferometer and applying DC voltages, a *V**_p_**L**_p_* of 1.86 V·mm was obtained and the extracted phase shifter propagation loss was 6 dB/mm. The response time for this phase shifter is around 8 ms. Xing et al. demonstrated a strip-loaded slot waveguide with a liquid crystal cladding phase shifter [111]. A better *V**_p_**L**_p_* of 0.0224 V·mm was achieved with the degradation of the phase shifter propagation loss to 10 dB/mm. The response time for this phase shifter was around 2 ms.

### 6.2. Phase Change Material

Phase change materials are a specific class of materials whose optical properties change significantly under external stimuli. Chalcogen-based alloys, especially Ge_2_Sb_2_Te_5_ (GST), attracts lots of attention and research interests due to its non-volatile nature [112,113].

The GST material will undergo a transition from an amorphous state to a crystalline state under external stimuli. The amorphous state of GST material could be highly transitive and electrically conductive. On the other hand, the crystalline state of GST material causes large optical absorption and is electrically resistive. The transition between these two states is generally achieved through heating, and optical or electric pulses usually act as external stimuli to heat the material. Furthermore, the GST material is widely used in all-optical photonic computing systems as the weight module due to its optical controllability and non-volatile nature. Some applications based on the GST materials are discussed in the Section 7.

## 7. Applications

As one of the most essential but important components, phase shifters play an important role in the development of reconfigurable PICs. Many high-performance reconfigurable devices based on phase shifters have been proposed, such as modulators [114,115,116], optical filters [117,118,119], and tunable delay lines [120,121]. In addition, an efficient phase shifter with low power consumption and high modulation speed paves the way to large-scale neuromorphic computing systems, photonic accelerators, optical phased arrays, on-chip spectrometers, and so on. In this section, we introduce several outstanding applications based on phase shifters.

### 7.1. Advanced Optical Computing Systems

In the post-Moore era, traditional computers based on the von Neumann architecture, which physically separates the computing module and the storage module, are facing speed and integration density bottlenecks. Many scientists began to explore the next generation of computing architectures to break though the limitations of Moore’s Law and demonstrated some promising computing platforms.

#### 7.1.1. Neuromorphic Computing System

The powerful computing capability and ultra-low power consumption of the human brain have attracted many scientists to reveal its mysterious working principle and mimic it using hardware. The development of micro- and nanofabrication technology and material science have made silicon PIC a promising platform for the physical imitation of the human brain, especially neural synapses.

The memory and learning mechanism of the human brain is based on the Hebbian learning rule. Action potentials (spikes) are generated by a neuron (pre-neuron) and propagate along the axon through a junction to the next neuron (post-neuron), which generates the postsynaptic action potentials. The junction is called a synapse, and the synaptic weight (w) determines the communication strength between the two neurons [122].

Cheng et al. proposed to use the PCM to simulate the synapse of nerve cells, as shown in Figure 10a [25]. Discrete PCM blocks were patterned on the taper waveguide to achieve adequate weight plasticity and easier control of the output state. By inputting different numbers of pulse signals, five states of the synaptic output were realized. Furthermore, an all-optical method was realized to modulate the synaptic weight.

Moreover, Feldmann et al. built a photonic neural network containing four neurons and sixty optical synapses based on spiking neurons, combining wavelength division multiplexing (WDM) and a PCM-based ring resonator to achieve weight addition and nonlinear activation (Figure 10b) [123]. Not only supervised learning but also the unsupervised learning training method can be realized through a feedback mechanism. They built an all-optical fully connected neural network that contains four neurons and successfully differentiated four 15-pixel images.

#### 7.1.2. Photonic Accelerator

Matrix multiplication is one of the most basic and important calculations in traditional computing architectures, especially in the field of neural networks and deep learning. In the process of deep learning, the weight matrix is fixed after training, and nonlinear operations are often performed. Considering that, all-optical computing could be a valuable solution for neural networks. In the all-optical neural network, the weights are implemented either by modulating the splitting ratio of the MZI through a phase modulator, or directly by changing the optical absorption rate of the PCM material. Shen et al. proposed an optical implementation of matrix multiplication using the MZI optical coherence module (Figure 10c) [124]. Before the signal was input into the optical neural networks (ONN), the authors preprocessed the input signals into a high-dimensional vector, and then encoded them into pulse signals of different amplitudes. Each layer of ONN contains an optical interference unit (OIU) to represent matrix multiplication and an optical nonlinearity unit (ONU) to implement nonlinear activation functions. In the experimental setup, the OIU is implemented by 56 MZIs, each containing a thermo-optics phase shifter. The function of the phase shifter is to change the splitting ratio of the MZI to route the optical signal and implement matrix multiplication. One more thermo-optics phase shifter was patterned on the output of the MZI to control the differential output phase. The authors then built an ONN containing four layers of OIUs with four neurons in each layer and showed acceptable performance (76.7% accuracy) in vowel recognition.

In addition, Feldmann et al. realized the parallel computing of matrix multiplication by combining PCM and optical frequency, which greatly improved the operation speed (Figure 10d) [36]. Zhang et al. utilized the MZI coherence and achieved complex-value calculation through optical neural networks (Figure 10e) [125]. Some promising works such as logic gate realization, classification tasks, and handwriting recognition were proposed.

### 7.2. Optical Phased Array

Inspired by array radars in electronics, the optical phased array has developed rapidly in the past two decades. OPAs have become a convincing candidate for optical communication in free space, LiDAR mapping, and spatially resolved optical sensors, benefitting from its precise and flexible steering angle of emitted light. Generally, OPAs are composed of an incident light coupler, phase shifter array, and grating emitters. Two-dimensional steering angles can currently be achieved, where one steering angle is controlled by the wavelength of the input light, and the other direction is controlled by the phase shifter. Considering the large-scale and densely integrated on-chip optical circuits, a phase shifter with high efficiency, low phase noise, and low power consumption is needed. Thermo-optics phase shifters are mainly used in OPA systems due to their easy access from commercial foundries and small footprint. Hutchison et al. achieved an ultra-high-resolution phase array by carefully designing a non-uniform emitter spacing, which showed 80° steering in the phased-array axis and 0.14° divergence with over 500 resolvable spots [27]. Sun et al. also achieved an 8 × 8 active phased array using directional couplers with different coupling ratios to obtain equal power emitting (Figure 11a) [29]. The thermo-optics phase shifters with doping silicon heaters are used to actively tune the phase in horizontal and vertical directions.

### 7.3. Multi-Functional Signal Processing Systems

Inspired by FPGAs in the field of electronics, Perez et al. proposed a hexagonal mesh structure in which each side of the hexagon has a phase shifter enabling a particularly large number of functions as shown in Figure 11b [126], such as single-input/single-output FIR filters, optical ring resonators, coupler resonator waveguides, side-coupler integrated spaced sequences of optical resonators, ring-loaded MZIs, and so on. The structures greatly improve the scalability and functionality of photonic integrated circuits.

### 7.4. On-Chip Spectrometer

The spectrometer is currently an important calibration and measurement tool in industry and laboratories. Although current bulky spectrometers can achieve high-resolution measurements, spectrometers currently have a trend towards miniaturization, and researchers have made great efforts in this regard [128,129,130,131]. The integrated phase shifters offer the on-chip light splitting and routing functions, which could enable the spectrometer application by creating on-chip light interference. Kita et al. demonstrated a digitalized Fourier transform (FT) spectrometer using the silicon PIC chip as shown in Figure 11c [127]. By constructing the optical switch with phase shifters, a tunable optical path difference was realized, controlling the thermo-optics phase shifters. The miniaturized FT spectrometer obtained a high resolution and scalability features through combining with machine learning regularization techniques, which achieved significant resolution enhancement beyond the classical Rayleigh criterion. As thermo-optics phase shifters are easily accessible in the silicon photonic foundry, the authors took the foundry service for the well-packaged chip device for the experimental demonstration.

## 8. Discussion

In this paper, we review the modulation mechanisms, optimized structures, and the performance of MEMS, thermo-optics, and free-carrier-depletion-based phase shifters. Trade-off between each FOM is the key in designing individual devices and selecting an appropriate phase shifter in a complicated system. It is hard to improve all FOMs simultaneously. The mechanical dimensions of the MEMS actuator have opposite effects on the applied voltage and modulation speed, while the initial position of the MEMS actuator affects the (dynamic) insertion loss and modulation efficiency. For thermo-optics phase shifters, a balance between the modulation efficiency and modulation speed needs to be determined according to the applications, and footprint and thermal crosstalk are sometimes important considerations. For free-carrier-depletion-based phase shifters, the free-carrier concentration and distribution affect the modulation efficiency, insertion loss, and modulation speed simultaneously.

On the other hand, these three kinds of phase shifters complement each other from the perspective of the application. The free-carrier-dispersion-based phase shifter is widely used in applications requiring high-speed phase modulation, such as telecommunications. However, it has the inherent disadvantage of relatively large dynamic insertion loss. The thermo-optics-based phase shifters offer efficient and stable phase modulation without dynamic insertion loss. However, the layout of the thermo-optics phase shifters must be carefully designed in large-scale PICs due to the limitation of large power consumption and thermal crosstalk. MEMS-based phase shifters appeared around two decades ago and had major developments in the past five years. Benefitting from its extremely low power consumption and no thermal crosstalk, MEMS-based phase shifters show significant potential for the future dense PIC applications. Nevertheless, due to the fatigue and other failure risks of non-solid-state systems, the packaging and long-term stability of MEMS-based phase shifters are still worth investigating.

In the future, in addition to the improvement of modulation efficiency and insertion loss, the dense integration and commercialization of silicon photonic phase shifters need further investigation, including the reduction in power consumption and footprint and the optimization of the packaging technologies, to name a few. Moreover, due to their excellent optical and electro-optics properties, heterogeneous integrated materials (Ge-on-Si, graphene, LiNbO_3_, etc.) have attracted great interest and flourished.

## Figures and Tables

**Figure 1 micromachines-13-01509-f001:**
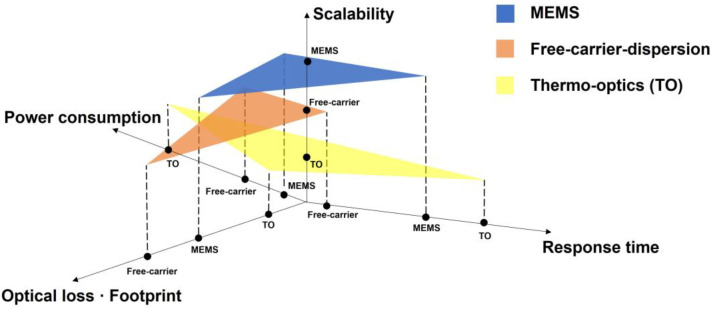
Semi-quantitative comparison between available methods for silicon photonic phase shifters.

**Figure 2 micromachines-13-01509-f002:**
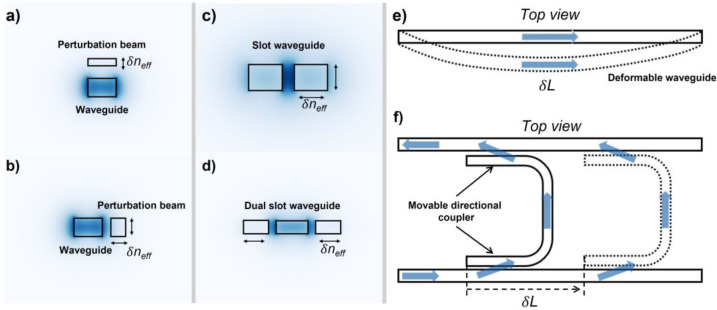
MEMS phase shifters based on evanescent field perturbation by (**a**) a silicon beam above the bus waveguide, (**b**) a cutoff width silicon beam is next to the bust waveguide; confined optical mode modulation through (**c**) slot waveguide, (**d**) dual slot waveguide; and modulating optical path length through (**e**) deformable waveguide, (**f**) horizontal directional coupler.

**Figure 3 micromachines-13-01509-f003:**
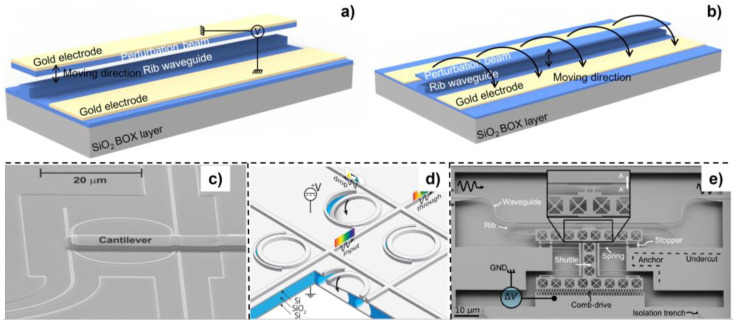
MEMS phase shifter through evanescent field perturbation utilizing (**a**) a silicon nitride beam coated with Au above the bus waveguide, (**b**) a pure silicon nitride beam actuated by gradient electric field force above the bus waveguide, (**c**) a silicon cantilever above the ring resonator (reprinted with permission from [54] © The Optical Society), (**d**) tunable ring resonator on the SOI (reprinted with permission from [55] © The Optical Society), (**e**) in-plane motion silicon beam perturbation (reprinted with permission from [45] © The Optical Society).

**Figure 4 micromachines-13-01509-f004:**
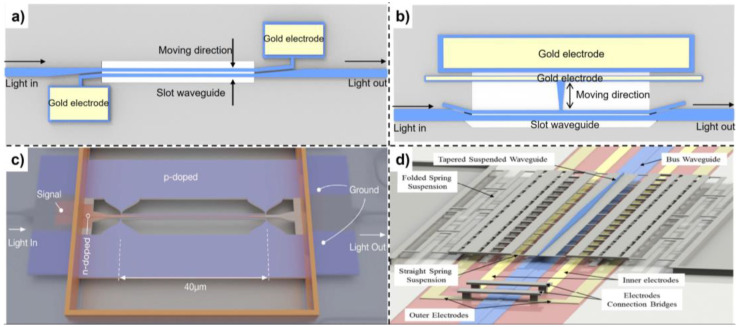
MEMS phase shifter through confined optical mode modulation utilizing (**a**) symmetric slot waveguide, (**b**) asymmetric slot waveguide, (**c**) dual slot waveguide (reprinted with permission from [59] © The Optical Society), (**d**) vertical directional coupler (reprinted with permission from [60] © The Optical Society).

**Figure 5 micromachines-13-01509-f005:**
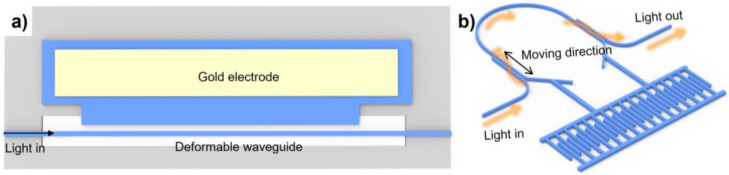
MEMS phase shifter through modulating optical path length utilizing (**a**) deformable silicon strip waveguide, (**b**) a pair of horizontal directional couplers.

**Figure 6 micromachines-13-01509-f006:**
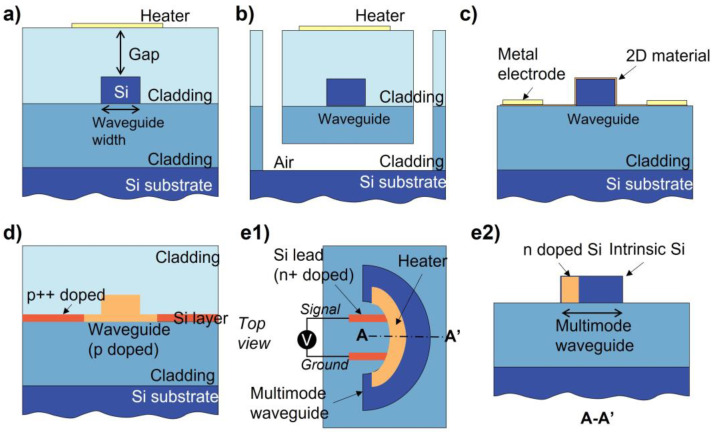
Various structures of thermo-optics phase shifters. (**a**) Traditional thermo-optics phase shifter, (**b**) thermo-optics phase shifter with air trench, (**c**) thermo-optics phase shifter with 2D material heaters, (**d**) thermo-optics phase shifter with doping silicon heater, (**e1**) bended thermo-optics phase shifter with doping silicon heater and (**e2**) its cross-sectional schematic.

**Figure 7 micromachines-13-01509-f007:**
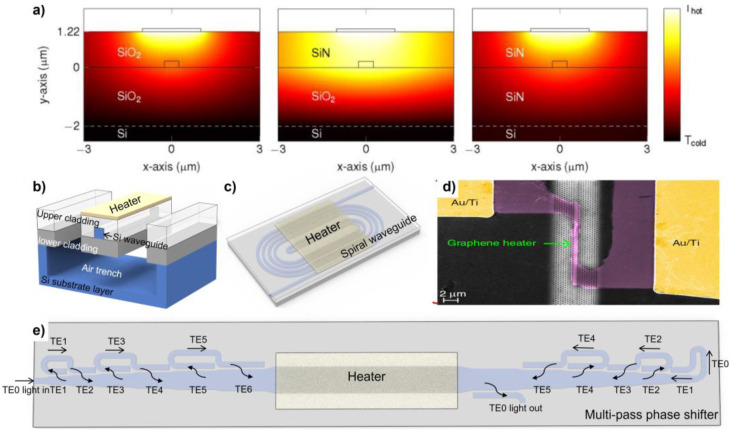
(**a**) Steady-state heat distribution for conventional thermo-optics phase shifters with different claddings (reprinted with permission from [68] © The Optical Society.), (**b**) free-standing thermo-optics phase shifter, (**c**) spiral waveguide thermo-optics phase shifter, (**d**) slow-light-enhanced thermo-optics phase shifter with graphene heater (reprinted with permission from [69] © 2017 Spring Nature), (**e**) multi-pass structure-based thermo-optics phase shifter.

**Figure 8 micromachines-13-01509-f008:**
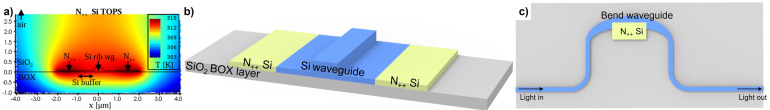
(**a**) Steady-state heat distribution for rib waveguide with doping silicon heaters, (**b**) rib waveguide thermo-optics phase shifter, (**c**) bended thermo-optics phase shifters with doping silicon heaters embedded in an MZI.

**Figure 9 micromachines-13-01509-f009:**
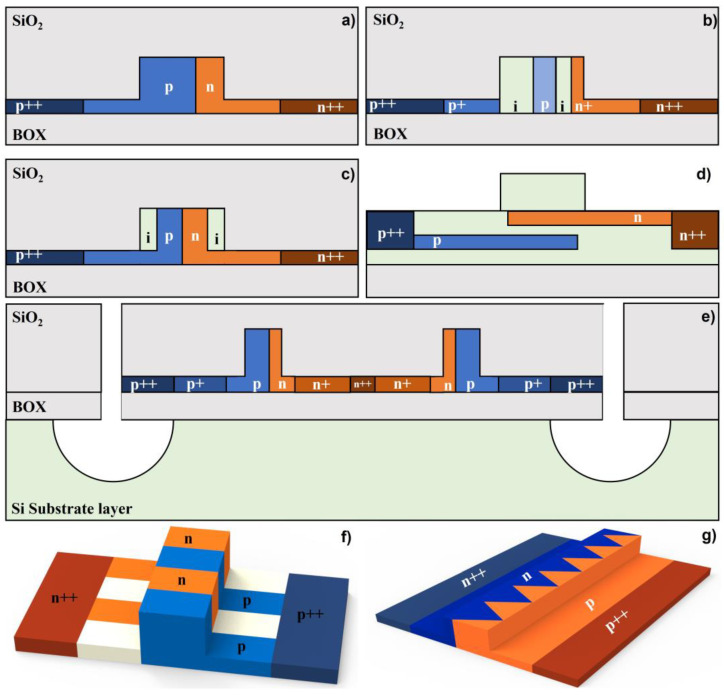
Various structures of free-carrier-depletion-based phase shifters. (**a**) Phase shifter with offset carrier doping, (**b**) PIPIN phase shifters, (**c**) phase shifter with counter doping at corners, (**d**) phase shifter with epitaxy fabrication, (**e**) phase shifters with substrate removement, (**f**) interleaved structure phase shifter, (**g**) zig-zag structure phase shifter.

**Figure 10 micromachines-13-01509-f010:**
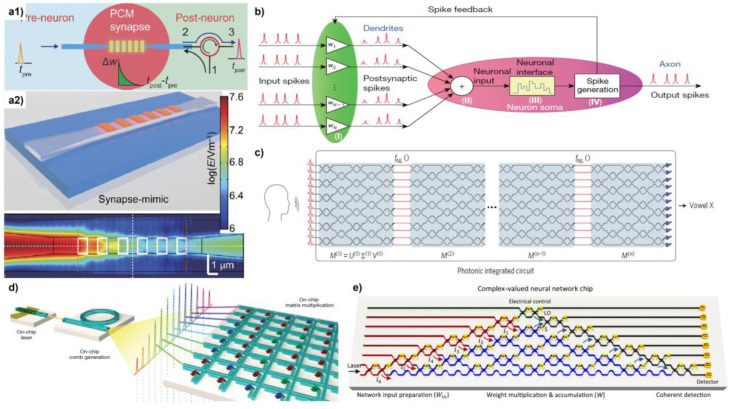
(**a1**) Schematic of the integrated photonic synapse. (**a2**) Top: Schematic of the photonic synapse realized by six discrete GST islands on the taper waveguide. Bottom: E-field distribution with all GST islands in crystalline states (reprinted with permission from [25] © 2017 AAAS.). (**b**) Schematics of the all-optical spiking neuronal circuits (reprinted with permission from [123] © 2019 Spring Nature). (**c**) All-optical fully integrated coherent nanophotonic network (reprinted with permission from [124] © 2017 Spring Nature). (**d**) Schematic of a parallel convolutional processing photonic architecture (reprinted with permission from [36] © 2021 Spring Nature). (**e**) Schematic of the optical neural chip in implementing complex-valued networks (reprinted with permission from [125] © 2021 Spring Nature).

**Figure 11 micromachines-13-01509-f011:**
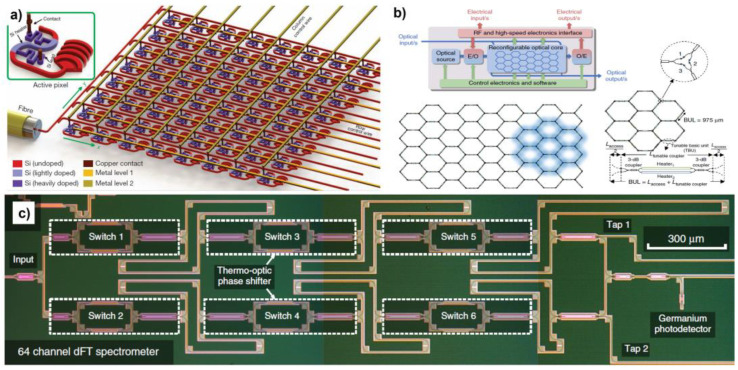
(**a**) Schematic of an 8 × 8 active phased array utilizing thermo-optics phase shifter (reprinted with permission from [29] © 2013 Spring Nature). (**b**) Schematic of a mesh structure multipurpose signal processor core (reprinted with permission from [126] © 2017 Spring Nature). (**c**) An optical image of an on-chip digital Fourier transform spectrometer (reprinted with permission from [127] © 2018 Spring Nature).

**Table 1 micromachines-13-01509-t001:** Performance summary of MEMS-based phase shifters.

Modulation Mechanism	*V_π_**⋅L_π_* (V·mm)	Insertion Loss (dB)	Modulation Speed (MHz)	Ref.
Evanescent field perturbation	0.432	0.5	2.3	[53]
	1.7	0.5	0.58	[56]
	0.535	0.33	0.503	[45]
Confinement optical mode modification	0.02	0.04	0.26	[59]
	0.588	5	1.1	[61]
	0.84	0.47	1.177	[58]
Optical path length adjustment	75	0.1	0.139	[62]
	-	0.4	0.153	[63]

**Table 3 micromachines-13-01509-t003:** Performance summary of free-carrier-depletion-based phase shifters.

Doping Distribution	P Conc. (cm^−3^)	N Conc. (cm^−3^)	Modulation Efficiency (V·cm)	Propagation Loss (dB/mm)	Modulation Speed (GHz)	**Ref.**
PN junction in the center of the waveguide	5 × 10^17^	5 × 10^17^	1.9 (3V)	1.2	>20	[91]
	7 × 10^17^	5 × 10^17^	3.5 (3V)	1	10	[92]
	1 × 10^18^	3 × 10^18^	1.59	3.2	27	[93]
	2 × 10^17^	2 × 10^17^	1.7 (3V)	1.2	12	[94]
	2 × 10^18^	2 × 10^18^	1.2 (3V)	4.5	4.3	[95]
	4 × 10^17^	1.3 × 10^17^	3.2 (0–4V)	1	46	[96]
Offset PN junction	2 × 10^18^	3 × 10^17^	14.3	-	8	[97]
	2 × 10^17^	6 × 10^17^	11 (3V)	-	19	[98]
	2 × 10^17^	2 × 10^17^	1.8 (3V)	1.6	27.8	[99]
	3 × 10^17^	1.5 × 10^18^	2.8 (4V)	5	40	[100]
Interleaved waveguide	2 × 10^17^	2 × 10^17^	1.7 (3V)	1	20	[101]
	2 × 10^17^	2 × 10^17^	1.4 (3V)	1.7	11.8	[102]
	2 × 10^18^	2 × 10^18^	0.8 (4V)	3.5	12.6	[95]
	5 × 10^17^	1 × 10^18^	2.4	2.1	20	[89]
Zig-zag waveguide	2 × 10^17^	4 × 10^17^	-	-	23	[90]
PIPIN junction	8 × 10^17^/ 3 × 10^17^	1 × 10^18^	3.5	1	40	[85]
Corner doping concentration	-	-	2.67 (−6V)	-	8.9	[86]
Wrapped PN junction	-	-	0.52 (2V)	-	50	[103]

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
