# Peer review of "Silicon Photonic Phase Shifters and Their Applications: A Review"

_micromachines, 2022, doi:10.3390/mi13091509_

Round 1
Reviewer 1 Report
The authors give a broad overview of the current progress in MEMS optical phase shifters and thoroughly explain their pros and cons comparing with modulators of other types (thermo-optical, plasma dispersion, Pockels effect-based). I recommend the article for publication after couple comments are addressed.
1) Page 4, paragraph 3.2.1. Oxide-to-air transition is mentioned as one of IL sources for MEMS modulators. I think it will be good to add that MEMS modulators driven by piezoelectric actuation do not suffer from this, as waveguide is still fully oxide-cladded, and neff is changed through elasto-optic effect. Examples references are available from Lionix International and Ghent university papers
(https://opg.optica.org/oe/fulltext.cfm?uri=oe-23-11-14018
https://opg.optica.org/abstract.cfm?uri=cleo_at-2017-JTh5C.7).
2) Page 15, Table 3. I think authors should add a work describing a wrapped PN junction modulator, as it demonstrates record-low VpiL of 0.59 V.cm.
(https://ieeexplore.ieee.org/document/8510799)
3) It would be nice to see a small discussion on reliability of MEMS phase shifters. As for any non-solid-state system, moving parts of MEMS modulators possess and increased risk of failure. Will be good to see examples where MEMS modulators demonstrate long lifetimes.
Author Response
Please see the attachement.

Reviewer 2 Report
The Authors review different types of silicon photonic phase shifters, including MEMS, thermo-optics and free-carrier depletion types. The manuscript is well written. Here, my comments to the manuscript:
- The Authors should justify the choice of just 3 techniques, neglecting other ones, as magneto- or electro-optics. As example, electro-optic effect via graphene capacitor has been widely used, aiming at achieving very low response time (see, e.g., Design of an ultra-compact graphene-based integrated microphotonic tunable delay line. Optics express, 26(4), 4593-4604, 2018; Double-layer graphene optical modulator, Nano Lett. 12(3), 1482–1485, 2012; Design optimization of single and double layer Graphene phase modulators in SOI, Opt. Express 23(5), 6478–6490, 2015; Silicon graphene Bragg gratings,” Opt. Express 22(5), 5283–5290, 2014).
- In the Section 3 and 4, the Authors should discuss on the material used for the tuning (i.e. gold, TiN, …), highlighting the main advantages and disadvantages, to address the reader to choice the best material for the target application.
- A discussion section should be inserted, comparing all the discussed techniques in terms of fabrication, performance and future trends.
- The applications section should be enlarged. Modulators (e.g. Graphene–silicon phase modulators with gigahertz bandwidth. Nature Photonics, 12(1), 40-44, 2018; Micrometre-scale silicon electro-optic modulator. nature, 435(7040), 325-327, 2005), filters (e.g. High performance and tunable optical pump-rejection filter for quantum photonic systems. Optics & Laser Technology, 139, 106978 , 2021; Highly tunable photonic crystal filter for the terahertz range. Optics letters, 30(5), 549-551, 2005), delay lines (e.g. Silicon graphene reconfigurable CROWS and SCISSORS,” IEEE Photonics J. 7(2), 1–9, 2015, “Photonic crystal tunable slow light device integrated with multi-heaters,” Appl. Phys. Lett. 100(22), 221110, 2012) and so on should be inserted.
Round 2
Reviewer 2 Report
The Authors have properly modified the manuscript according to the Reviewer suggestions. Therefore, I suggest the manuscript publication.